# Endangered Forest Communities in Central Europe: Mapping Current and Potential Distributions of Euro-Siberian Steppic Woods with *Quercus* spp. in South Slovak Basin

**DOI:** 10.3390/biology12070910

**Published:** 2023-06-25

**Authors:** Peter Oravec, Lukáš Wittlinger, František Máliš

**Affiliations:** 1Protected Landscape Area Cerová Vrchovina, State Nature Conservancy of the Slovak Republic, Železničná 31, 979 01 Rimavská Sobota, Slovakia; peter.oravec@sopsr.sk; 2Department of Ecology and Environmental Sciences, Faculty of Natural Sciences and Informatics, Constantine the Philosopher University, Tr. A. Hlinku 1, 949 74 Nitra, Slovakia; 3Department of Phytology, Faculty of Forestry, Technical University in Zvolen, T. G. Masaryka 24, 960 01 Zvolen, Slovakia; frantisek.malis@tuzvo.sk

**Keywords:** oak forests, habitat code 91I0*, territory code SKUEV0957 uderinky, natura 2000, South Slovak Basin

## Abstract

**Simple Summary:**

Temperate deciduous forests are among the endangered forest communities in Central Europe currently facing several anthropogenic impacts, such as the intensification of forest management (including global climate change), but also the consequences of natural disturbances on plant diversity and the dynamics of natural forest communities. If we want to preserve the natural composition of forest ecosystems, it is necessary to understand the changes that significantly negatively affect complex biological systems. As part of the research, we focused on the evaluation and identification of the priority habitat 91I0* Euro-Siberian steppic woods with *Quercus* spp., which is classified as an endangered habitat. The aim of this research was to supplement the current list with new habitat sites of European importance at the level of the South Slovak basin. The methodology was based on phytosociological research, and it was a syntaxonomic classification of forest communities with a special focus on Euro-Siberian oak forests on loess and sand. The results of the current and potential expansion of the habitat of the Euro-Siberian oak forests on loess and sand in the South Slovak basin will help to meet the national and legislative requirements of the European Commission under the Directive on the status of habitats and species of European importance.

**Abstract:**

In this article we focus on the issue of determining the presence and status of the priority habitat 91I0* Euro-Siberian steppic woods with *Quercus* spp. in the South Slovak basin. As part of the issue, we try to verify the correctness of the procedure of the State Nature Conservancy of the Slovak Republic in the search for potential habitats and areas of European importance by converting the typological map to a map of habitats. Habitat 91I0* occurs in Slovakia in the form of three subtypes, namely Thermophilous and supra-Mediterranean oak woods (*Carpineto-Quercetum* and *Betuleto-Quercetum*), Acidophilous oak forests (*Quercetum*), while the last-named subtype is divided into two subunits: Medio-European acidophilous oak forests—part A and Pannonic hairy greenweed sessile oak woods—part B. Due to the current unsatisfactory state of the mentioned habitats, the requirement of the State Nature Conservancy of the Slovak Republic is to find and add new areas with the occurrence of habitat 91I0* in the south of Central Slovakia. During the mapping in the Lučenecká and Rimavská basins, greater emphasis was placed on the occurrence of the subtype Thermophilic Pontic-Pannonian oak forests on loess and sand, but its presence has not been confirmed. Subsequently, we focused on the search and identification of habitats in the model area, which is the area of European importance SKUEV0957 Uderinky. The result is a map of habitats in this area, which we then compare with a typological map, which determines the reliability of the converter used by the State Nature Conservancy of the Slovak Republic.

## 1. Introduction

The Biome of deciduous broad-leaved forests is characterized by Central European oak groves and beeches, which are the basic type of ecosystems found in the lowlands, hills, and highlands. Forests have become part of the socio-economic activities of man, which brings a negative impact on the original European habitats, which have been preserved in relatively unaffected parts of Slovak nature. Natural forests are considered a reservoir of great biological diversity constituting one of the most important ecosystems in Europe. *Quercus* study is essential to assess the ecological conservation of forests and is of economic importance for different industries [1]. The intensification of forestry and the change in the way forests are managed is not the only factor that significantly affects the spatial-area boundary of the expansion of forest ecosystems. Currently, the limiting factors include climate change, which is gradually leading to a decline in biodiversity, which are also interconnected, and their solution is key to sustaining life on Earth. From this point of view, there is a need to place greater emphasis on the protection of nature, so that we can ensure optimal conditions for the preservation of the original European phytocenoses and zoocenoses [2,3]. The conversion of forests from complex natural ecosystems to simplified commercial woodlands is one of the major causes of biodiversity loss. To maintain biodiversity, we need to understand how current management practices influence forest ecosystems. We studied the effects of forest successional stages and management intensity on the abundance and species richness [4]. Similar studies have been conducted in the territory of the Slovak Republic with a focus on damage to soil and remaining forest stands by machines, evaluation of abiotic controls of wind disturbance using the generalized additive model [5,6,7], with a focus on spruce forests [8,9], and an analysis of the qualitative features of beech and oak trunks as a determinant of quality assessment [10].

Xero-thermophile oak woods of the plains of southeastern Europe are in a climate that is very continental, with high changes in temperature. The substrate consists of ‘Loess’ (chernozem soils). *Quercus robur, Quercus cerris*, and *Quercus pubescens* dominate in the treelayer of this habitat type, which is rich in continental steppic vegetation elements and geophytes of the *Aceri tatarici-Quercion* Zólyomi 1957 [11]. The oak stands we deal with in our work and the overall plant communities associated with them are currently among the most endangered. They are particularly threatened by invasive plant and tree species, climate change, and forestry changes. Oak (*Quercus* spp.), as a strong tree, is currently undergrown with hornbeam (*Carpinus* spp.), which belongs to the shady trees, or is replaced by acacia (*Robinia* spp.). The enrichment of the soil with nutrients and nitrogen is related to the change in management. In the past, livestock grazing and litter digging were often carried out on them, with the sprout system of management predominating. The soil was poorer in nutrients, which suited these communities. These stands also include the 91I0 Euro-Siberian steppic woods with *Quercus* spp., which we have researched, which has its northwestern border at the expansion area in our country. At present, the habitat is highly endangered [12,13].

Based on a request from the European Commission, the State Nature Conservancy of the Slovak Republic (hereinafter referred to as the SNC SR) initiated the process of finding and supplementing the 91I0* habitat in Slovakia, due to insufficient connectivity between southwestern and southeastern Slovakia. 

In the area of southern Central Slovakia, especially in the South Slovak basin, it was mainly the occurrence of Ls3.2 Tartar maple steppe oak woods. The subject of reconnaissance were also other habitats belonging to habitat 91I0*, namely Ls3.3 White cinquefoil oak woods and Ls3.5.2—part B, Pannonic hairy greenweed sessile oak woods.

Priority habitats of European importance, which include habitat 91I0*, should have up to 60% of their area of occurrence in territories of European importance. In Slovakia, it is currently only 20%. Therefore, Slovakia does not meet the important prerequisites for creating suitable conditions for the preservation of rare species. These prerequisites are the sufficient size of habitats, their quality, and their connectivity.

For this issue, we focus on approaches to the classification of this vegetation, the history of oak forests, the state of the habitat, and its geographical distribution in the Slovak Republic. We also focus on verifying the correctness of the typological map converter to the habitat map used by SNC SR. The evaluation of the adequacy of the converter is carried out in the model area SKUEV0957 Uderinky with the occurrence of the priority habitat 91I0*. By analyzing these data, we aim to address the following questions: (i) whether the approaches to the classification of vegetation are correct within the converter of the typological map with the occurrence of the priority habitat 91I0* Euro-Siberian steppic woods with *Quercus* spp., (ii) in which localities of the study area is the priority habitat represented, and (iii) what is the overall assessment of the state of habitat protection of European importance in Slovakia.

The results of the study should be used in the assessment of environmental impacts in connection with land use change, in the implementation of landscape–ecological principles in spatial planning processes, and in the fulfillment of national legislative requirements and requirements of the European Commission’s directives on the state of habitats and species of European importance.

## 2. Materials and Methods

### 2.1. Study Area

The South Slovak basin is a landscape complex in the Lučensko-košická depressions extending into the Ipeľ and Slaná river basins. It is bordered in the north by the Krupinská plain and Slovenské Rudohorie, in the east by the plains of the Slovak Karst, and in the south by the system of volcanic mountains Börzsöny and Cserhát in Hungary, and the Cerová highlands in Slovakia (Figure 1). Within the regional geological division of the Western Carpathians [14], the South Slovak basin is formed by the Rimavská Basin, the Cerová highlands, the Lučenecká, and Ipeľská basins. These basins are filled with medium-miocene andesite volcanoclastics and low-thickness upper miocene sediments. The area is represented by rock complexes that belong to the tectonic units veporik, gemerik, turnaik, and silicik. The Rimavská and Lučenec basins are built mainly by fluvial sediments in the representation of lithofacial unstructured alluvial clays, or sandy to gravelly clays of the valley floodplains and the floodplains of mountain streams [15].

The studied area belongs to the Alpine–Himalayan system, the Carpathian sub-system from a geographical point of view, and it is divided into the province of the Western Carpathians and the Lučensko-košická depressions [16]. Furthermore, its area extends into the whole of The South Slovak basin and the subunits of the Rimavská basin and Lučenecká basin. Based on the climatic–geographical types of the Slovak Republic, in terms of climatic conditions, the affected area belongs to a warm climatic area, and the districts are warm and moderately warm [17]. The current vegetation is significantly affected by climate change, which affects forest management mainly at lower altitudes—alternating between extreme dry and warm days. In the Slovak Republic, in the period between 1881–2017, there was a significant increase in annual air temperature by 2.0 °C. Accordingly, there was a slight trend in annual rainfall with a total of about 1% on average. Annual precipitation totals have fallen by more than 10% in the south of the country. Relative air humidity decreased up to 5% in the south-west of the Slovak Republic. There is evidence of gradual desertification, particularly in the south of the country (increase in potential evapotranspiration and decrease in soil moisture), nevertheless, the year 2010 and the cold half-year of 2012/2013 were the wettest since 1881. Significant increases in regional floods and flash floods were recorded after 1993. The temperature increase in the last 38 years is even more significant than in the whole 1881–2017 period and the precipitation totals have started to increase slightly since 1994 (the extreme year of 2010 is an exception) [18,19].

The area belongs to the Ipeľ river basin. The river network of this basin consists of left-hand tributaries: the Uhorský stream, Poltárica, Petrovský stream, Suchá, Babský stream, Mučínsky stream, and Kamenec, as well as right-hand tributaries: the Chocholná, Banský stream, Krivánsky stream, Mašková, Tisovník, Stracinský stream, Krtíš, Čebovský stream, Olvár, Krupinica, Štiavnica, Búr, and Jelšovka.

They are typical for the investigated territory based on the geological bedrockPseudogleys, Fluvisols, Luvisols, Mollisol, Cambisols, Pararendzins, and Regosols [20].

In the area there are loamy-sandy, sandy-loamy, loamy, clayey-loamy, and clayey soil species [21]. In terms of the phytogeographical division of Europe, the area is part of the Holarktis area, the Eurosiberian sub-region, and the province of Central Europe [22]. From the phytogeographical–vegetation division, the area belongs to two subdistricts of The South Slovak basin: the Lučenecká basin and the Rimavská basin [23].

### 2.2. Data Analysis of Functional Characteristics of Plant Communities

During the vegetation period from 2018–2022 (1 March–31 October), we carried out field research. When conducting the field research, we used a chorological approach, which created a joint analysis of the habitat location and topographical set, in which we focused on creating an inventory of taxa at the sites [24]. The collection and synthetic processing of information about vegetation and their abiotic environment directly in the field served to create a biogeographical regionalization of the studied area. In terms of time, it was semi-stationary field research (repeated, but not continuous measurement of the dynamic characteristics of the landscape, while we focused on the dynamics of the plant communities) [25]. We used the following categories for evaluation: representation of trees, herbs, and shrubs; age structure; rejuvenation; vertical construction of a forest habitat; thick and particularly valuable trees; thick dead wood; health status; and negative impact of the surroundings.

As part of the monitoring, we focused on the main habitat 91I0* Euro-Siberian steppic woods with *Quercus* spp. In dry and warm areas of the South Slovak basin. On the territory of the Slovak Republic, habitat 91I0* is classified into three subtypes: Ls3.2 Thermophilous and supra-Mediterranean oak woods (*Carpineto-Quercetum*), Ls3.3 Thermophilous and supra-Mediterranean oak woods (*Betuleto-Quercetum*) and a Ls3.5 Acidophilous oak forests (*Quercetum*), while the last-named subtype is divided into two subunits—stunted woodpeckers with *Genista ilosa* (Ls3.5.2—part B, Pannonic hairy greenweed sessile oak woods), which belong to Natura 2000, and other types of acidophilic oak forests (Ls3.5.1—part A, Medio-European acidophilous oak forests), which are only a habitat of national importance [12]. At each locality, a map of habitats (software: ArcGIS, ESRI ArcView GIS) was created by field mapping at the level of classification according to the work of Stanová and Valachovič [12]. Occurrence of habitats belonging to habitat 91I0* (Ls3.2, Ls3.3, and Ls3.5.2—part B) is defined as the estimated percentage of forest spatial distribution units (FSDU).

We present the scientific names of the taxa according to the work of [26]. We present summary information on the monitoring of habitats of European importance in the Slovak Republic based on the monitoring results of [27]. The criteria for assessing the conditions are defined by the monitoring methodology prepared for each habitat separately by [28]. We provide a biogeographic assessment of the state of habitat protection in accordance with Council Directive 92/43/EEC. of 21 May 1992, on the protection of natural habitats and wild animals and plants from a publicly available database [11,29].

## 3. Results

We have divided the results of the work into three subsections. In the first chapter, we generally analyze the current state of the habitat within selected countries of the European Union. In the second chapter, we define the distribution of habitat 91I0* in the territory of the Slovak Republic, the state of protection, and current management (Figure 2). In the third chapter, eight locations that serve as model territories are analyzed in detail.

### 3.1. Habitat Assessments at EU Biogeographical Level

At the European Union biogeographical level, habitat 91I0* is found in five bioregions, namely the Pannonian, Alpine, Continental, Steppic, and Black Sea bioregions, and is present in seven countries (Bulgaria, Slovakia, Austria, Czech Republic, Poland, Romania, and Hungary).

In most areas, the habitat is in an unfavourable state with a deteriorating trend of development, which is mainly due to the poor current state of conservation. The largest area, and also the best structure of the habitat, is located in the continental and steppic bioregions, especially in Bulgaria and Romania (Table 1).

In the territory of Romania in the Pannonian and Continental bioregions, the status of habitat 91I0* is favourable, and forecasts estimate that this status will persist in the future due to a good current conservation status. The habitat is also in a favourable state in the territory of Bulgaria in all bioregions, however, here in the future its condition is expected to deteriorate to an unfavourable level due to the deteriorating state of conservation.

On the territory of Austria, the continental bioregion habitat is in favorable condition. The outlook for the future is getting worse.

In the territory of Slovakia in the Alpine and Pannonian bioregions, the state of the habitat has been unfavorable for a long time and the persistence of this state is also expected for the future. The same applies to localities in the territory of Poland located in the continental bioregion, to localities in the Czech Republic in the Pannonian and continental bioregions, and to localities in Romania in the steppic bioregion.

In the territory of Hungary in the Pannonian habitat, habitat 91I0* has been in poor condition for a long time and is not expected to improve in the future (Figure 3).

### 3.2. Protection Status and Distribution of Priority Habitat 91I0* in the Slovak Republic

Habitat 91I0*, in the form of its three subtypes, occurs mainly in the southern half of Slovakia (Figure 4). The group of Pontic-Pannonian oak forests (Ls3.2) has in our territory the northwestern border of its area with its original occurrence in the geomorphological units of the Borská lowland, Hronská upland, Ipeľská upland, Nitrianska upland, Podunajská plane, Východoslovenská plane, and Zemplínske highlands [13]. In the case of the group of White cinquefoil oak woods (Ls3.3), the original occurrence is reported in the Borská lowland, Hornád basin, Hornonitrianska basin, Ipeľ basin, Košice basin, Liptov basin, Lučenská basin, Pliešovská basin, Danube lowland, Poprad basin, Považského podolia, Rimavská basin, Spišsko-šarišského mezihoria, Štiavnické highlands, Turčianská basin, Východoslovenská lowland, Východoslovenská upland, Zvolenská basin, and Žilinská basin at altitudes from 150 to 700 m above sea level. Subtype Ls3.5.2—part B has an area of occurrence in the geomorphological units of the Little Carpathians, Považský Inovec, and Tribeč at altitudes from 250 to 500 m above sea level, while the occurrence has also been recorded in the area of European importance Uderinky, which is located in the geomorphological unit of the Revúcka highland and Southern Slovak basin.

At present, the habitat in the territory of the Slovak Republic is considered to be very endangered. Some subtypes have been preserved only in small areas; therefore, it is necessary to ensure the consistent protection of these remnants of thermophilous steppe oak stands [12].

The habitat is in poor condition in both regions in all three categories. The conservation status of the habitat in the Alpine bioregion represents up to 47.0% of the favourable status. In the Pannonian bioregion, it is 37.3%. In both cases, the unsatisfactory condition represents a value greater than 40.0%. The bad state is at 10,0% in the Alpine bioregion and 18.7% in the Pannonian bioregion (Figure 5). The habitat prospects in the bioregion are identical within a favourable state. The unfavorable state is three percentage points higher in the Alpine bioregion (49.0%). Poor condition is lower by 6.0%. In the Pannonian bioregion, and the unfavorable state is at a level of 59.0%. Poor condition is lower by 8.0%. The quality of the habitat is expressed at the level of 53% in the Alpine bioregion. Unsatisfactory conditions cover 37.0%. It is in a bad state of 10.0%. The quality of the habitat in the Pannonian bioregion is 45.3%. A total of 36.0% are in an unfavorable state. In poor condition up to 18.7%.

The landscape structure of the South Slovak Basin has been significantly influenced by human activity, with man influencing and shaping ecosystems and relief shapes since the Neolithic (6000 to 2900 BC.). The development of settlements, agriculture, and forestry had the greatest impact on the natural conditions of the environment. Changes in the use of the country were dynamically reflected in the overall result, as broad-leaved forests, pastures, vineyards, and landcover principally occupied by agriculture, with significant areas of natural vegetation, complex cultivation patterns, water bodies, sport and leisure facilities, industrial or commercial units, and discontinuous urban fabric [31].

The analysis of favourable, inadequate, and bad statuses should be based on anthropic factors that significantly contribute to the improvement or deterioration of habitats. The greatest positive impact on the state of habitat 91I0* in the Alpine bioregion is the restoration of the forest and its proper management (83.3%), while the most negative impact is the restoration of the forest with improper management (30%) and the construction of transport networks (24%), hunting and trapping of wild animals (22%) (damage to forest stands by animals—normalized and biologically tolerable state), abiotic natural processes (10%), invasive species (6%), changes in abiotic conditions (4%) and, last but not least, mining (1%) and soil pollution (1%) (Figure 6a,b). In the case of the Pannonian bioregion, the positive impact of forest restoration and proper management is even stronger than in the Alpine bioregion (90.9%), while the most negative impact is also the restoration of the forest with improper management (32%). In the Pannonian bioregion, the occurrence of invasive species poses a greater risk (26%). Just as in the Alpine bioregion, negative impacts are also caused by improper hunting management (18%), construction of transport networks (9%), and cultivation (3%) in the Pannonian bioregion. To the same extent, changes in the way of farming (3%) and the use of fertilizers (3%) also have a negative impact on the Pannonian bioregion (Figure 6c,d).

### 3.3. Evaluation of Partial Areas


**Ťahan (cadastral territory Sútor)**


The Ťahan locality is situated north of the village of Sútor in the Rimavská Sobota district. At the given locality, the dominant soil species is luvisol, but there is also pararendzina, which could condition the Ls3.2 habitat. However, the dominance of *Q. cerris* precludes this possibility. In the south-eastern part, the vegetation is illuminated, and the undergrowth is abundantly represented by shrubs. In this part of the stand there were mainly taxa such as *Quercus petraea*, *Q. robur*, *Q. cerris*, *Ligustrum vulgare*, *Seratula tinctoria*, *Brachypodium sylvaticum*, *Viola riviniana*, *Alliaria etiolate*, *Mycelis muralis*, *Tithymalus cyparissias*, *Astragalus glycyphyllos*, *Veronica chamaedrys*, *Poa angustifolia*, and *Mellitis melissophyllum*. The vegetation on the SW slope is diluted by reverse mining with a dense undergrowth of light-loving and nitrophilous species, e.g., *Humulus lupulus*, *Clematis vitalba*, *Sambucus nigra*, *Chelidonium majus*, *Geranium robertianum*, *Circaea lutetiana*, and *Robinia pseudoacacia* is also abundant. The base of the slope consists of a mixed stand of *Quercus* spp., *Pinus sylvestris*, *Carpinus betulus*, *R. pseudoacacia*, and *Acer tataricum*. The occurrence of the 91I0* habitat has not been confirmed at the given locality, but some parts of the stands can be classified as the 91G0 habitat, which is also a Natura 2000 priority habitat.


**Nad Opatovskou cestou (cadastral territory Rimavská Sobota)**


The stands at the site were included in the 91G0 (Ls2.2) habitat, but some parts have a transitional character to the 91I0* (Ls3.2, Ls3.3) habitat. This is indicated by the rare presence of forest-steppe species and species indicating seasonal wetting (*Betonica officinalis*, *Tithymalus olychrome*, *Lysimachia nummularia*, *Filipendula vulgaris*, *Anthericum ramosum, Pulmonaria mollis*, and *Deschampsia caespitosa*). *Frangula alnus* is relatively abundant in some parts. These are mostly wide flat ridges and gently sloping slopes on old river terraces. However, the most thermophilic species were present in the convex parts at the crossing points of plateaus and gentle slopes into the valley. At the same time, *Q. cerris* had a higher abundance in these places, approximately 40–60%, which contrasts with the inclusion in the 91I0* habitat. However, the vegetation in this locality is considerably ruderalized because of principal felling. In the past, livestock grazing was very likely. This assessment applies to most FSDU (364, 366, 368), especially to FSDU 358, where the share of 91I0* can be estimated at around 20%.


**Mestský les (cadastral territory Gemerček)**


This area has a hint of habitat 91I0* in the form of a transition between habitat Ls2.2 and Ls3.3 in FSDU 486, with a share of about 20%. However, *Quercus* spp. forming the main level, is undergrown mainly by *C. betulus*, because of which the herbal synthesis is very poorly developed until absent. In the southern part of the mapped area, this undergrowth of shading trees is not present and the layer of herbs is well developed, rich in species such as *P. angustifolia*, *Trifolium alpestre*, *Fragaria vesca*, *Vicia cassubica*, *Poa nemoralis*, *Vincetoxicum hirundinaria*, *Clinopodium vulgare*, *Lathyrus niger*, *Carex muricata* agg., *A. glycyphyllos*, *Crucia glabra*, *Viola hirta*, *Digitalis grandiflora*, *V. hirundinaria*, *A. ramosum*, *Festuca rupicola*, *Carex montana*, *Dactylis glomerata* agg., *Prunus spinosa*, and *Hieracium racemosum*.


**Kružno, local part Mlynárka**


The locality tends towards the Ls2.1 habitat, but according to the methodological guideline for mapping, Ls2.1 must have at least 30% representation of *Q. petraea*, which in this case does not indicate the studied locality. Most of the stand is undergrown with *C. betulus*, due to which the herbaceous floor is poorly developed. Completely absent in some places. The wood composition is dominated on the lower side of the slope mainly by *Q. robur* and *C. betulus*, while up the slope it is *Q. cerris*. Thus, colder variants of Ls2.2 and Ls3.4 occur at the site, probably also due to the underdeveloped vegetation shaded by *C. betulus*. On the lower side of the slope there is a mesotrophic–nitrophilic combination of *Pulmonaria obscura*, *Impatiens parviflora*, *Ajuga reptans*, *G. robertianum*, *Viola reichenbachiana*, *M. muralis*, *Geum urbanum*, *Stellaria holostea*, *Glechoma hirsute*, and *Urtica dioica*.


**Teriakovce**


The stand consists of *Q. petraea*, *Q. robus*, and *Q. cerris* and is undergrown with *C. betulus*. On the north-western slope there is the occurrence of mesophilic vegetation such as *V. riviniana*, *Scrophularia nodosa*, *Veronica officinalis*, *L. vulgare*, *Dryopteris filix-mas*, *Moehringia trinervia*, *Euonymus verrusosus*, *G. urbanum*, *M. muralis*, and *Fallopia convolvulus*. This part represents a kind of warm variant of Ls2.1. The other side of the valley, towards Maginhrad, is completely different from all other localities. Not Typical is the occurrence of loess and river sediments, but andesite. There is vegetation such as oaks on Central Slovak volcanics, but significantly more thermophilic. The upward slope begins the transition to forests—steppe with species such as *V. cassubica*, *Peucedanum cervaria*, *T. alpestre*, *P. nemoralis*, *H. racemosum*, *F. convolvulus*, *A. etiolate*, *Carex michelii*, *Sedum maximum*, *F. rupicola*, *C. vulgare*, *Luzula luzuloides*, *V. officinalis*, *Cruciata glabra*, *C. vulgare*, *T. cyparissias*, *V. chamaedrys*, *B. sylvaticum*, *Silene nutans*, *P. angustifolia*, *L. vulgare*, *Acer campestre*, *Verbascum austriacum*, *Sorbus torminalis*, and *A. ramosum*.


**Bottovo**


At the base of the slope in contact with the alluvium of the Belín brook is a clayey dark soil with a polyhedral structure where *Q. robur* dominates, but up the slope the soil changes to luvisol and gradually increases *Q. cerris*. This part can be included in Ls2.1, also based on a calibration meeting that took place in this location. In some places, the stand is considerably ruderalized in the representation of the species *U. dioica*, *Persicaria hydropiper*, and *I. parviflora*. Anthropophytes such as *Stenactis annua*, *Conyza canadensis*, and *Erechtites hieracifolius* occur in the illuminated places. On the flat top there are luvisols with a hint of gleystation, where *Q. robur* (90%) dominates with a rich undergrowth of trusses (mainly *Crataegus* spp.) and *A. tataricum* is also present. It occurs in the opposite stand based on the slope Ls2.2, and the change to forest with *Q. cerris* can be seen upwards. At the top there is again glazed luvisol, but *Q. robur* is only 20% dominated by *Carex brizoides*. Nevertheless, we classify the habitat as Ls3.4, a wet variant in the transition to Ls2.2. Other taxa present are *B. sylvaticum*, *V. hirundinaria*, *G. urbanum*, *C. glabra*, *F. convolvulus*, *C. vulgare*, *F. vesca*, *Agrostis stolonifera*, *L. nummularia*, *P. angustifolia*, *Hypericum hirsutum*, *T. cyparissias*. *A. tataricum* rejuvenates relatively well, *Crataegus* spp., and *L. vulgare*. From a phytopathological point of view, some parts of the stand can be described as significantly damaged by game. The presence of the 91I0* habitat has not been confirmed at the site, but there are 91G0 and 91M0 habitats, which are also subject to Natura 2000 protection.


**Bretka**


Despite the proximity of the Slovak Karst, there are still river sediments in the locality, gravelly stony terraces covered with loess. FSDU contains a species-rich hollow with a predominance of *Q. petraea*, which can be classified in the Ls3.3 (91I0*) habitat. The share of the Ls3.3 habitat in the stand can be estimated at 60%. However, there are few species indicating seasonal wetting, so the vegetation is transient to the Ls2.1 habitat. Representation of species is predominantly *Calamagrostis arundinacea*, *F. alnus*, *Dryopteris filix-mas*, *P. obscura*, *Trifolium medium*, *V. hirundinaria*, *P. angustifolia*, *Dryopteris filix-mas*, *Platanther bifolia*, *C. montana*, *F. vesca*, *A. reptans*, *B. sylvaticum*, *C. glabra*, *S. nodosa*, *V. officinalis*, *P. obscura*, *M. melissophyllum*, and *V. cassubica*. In stands (FSDU) 72, 36, 33c, and 34, there are gleystated habitats, with soil indicating eluvial processes (light top, deeper gravelly horizons of rusty color). Species more demanding on moisture and tolerating lower soil reaction are present. These parts could be classified as a Ls3.3 habitat, with a share of about 5–10% in these stands.


**Uderinky (cadastral territory Lovinobaňa—part Uderiná, Točnica)**


As part of field mapping, the habitat of European importance 91I0* in the Uderinky locality is found only in the form of subtype Ls3.5.2—part B. There was a total of three habitats at the site, namely the already mentioned habitat Ls3.5.2—part B, Ls3.5.1—part A, other types of acidophilic oak groves, and habitat Ls2.1 Carpathian oak-hornbeam forests (Figure 7). Floor E_0_ was very well-developed in these areas, but the species composition of floor E_1_ was relatively poor in species. The reason for the protection of this area is the preservation of the European important habitat 91I0*. The wood composition of this area is represented by *Q. petraea* agg., *C. betulus*, *F. alnus*, *Betula pendula*, *Fagus sylvatica*, *Q. cerris*, and *Cerasus avium*. The shrub floor is represented by *Rosa* spp., *Crataegus laevigata*, and *P. spinosa*. We distinguished habitat Ls3.5.2—part B from habitat Ls3.5.1—part A mainly according to the presence of the taxon *G. pilosa*, according to soil conditions, growth abilities of *Q. petraea* agg., and according to slope conditions. The eastern and south-eastern slopes were diametrically different, in their tree composition *C. betulus* were added to *Q. petraea* agg. and *B. pendula*. The soil in this area was deeper and richer in nutrients, which was also demonstrated by the occurrence of some mesophilic species, and the growth was more productive.

## 4. Discussion

Understanding the relationship between disturbance and forest community dynamics is a key factor in sustainable forest management and conservation planning [34]. A significant change of habitats is visible in the studied area. This is related to the long period of influence of oak sites in the past by frequent youngling of these stands, but also the intentional preference of oak. The structure of such stands is often altered, with a significant shortage of rough trees and rough dead wood. Due to predicted climate change, it is important to know to what extent trees and forests will be impacted by chronic and episodic drought stress, since oaks play an important role in European forestry [35]. For example, in Greece a total of 40% of the locations covered by deciduous oak forests were converted to agricultural areas, most of the rest of these locations were converted to vegetation-type characteristics of lower precipitation and soil fertility [36].

Within the habitat wood composition is relatively well preserved, but there are also invasive species (*R. pseudoacacia*), whose deliberate cultivation significantly affects the continued existence of this habitat. It is similar in the composition of the undergrowth, which consists of the original species, but there are also species of invasive character (*I. parviflora*) [3,13].

Tree species diversity is a key parameter to describe forest ecosystems. It is, for example, important for issues such as wildlife habitat modeling and close-to-nature forest management [37]. Based on a visit to model areas and a survey of selected localities, it is possible to state that the 91I0* habitat is in the Rimavská kotlina area only occur in the form of the Ls3.3 Thermophilous and supra-Mediterranean oak woods (*Betuleto-Quercetum*), which, however, also has only a marginal to transitional character to other habitats. Habitat Ls3.2 Thermophilous and supra-Mediterranean oak woods (*Carpineto-Quercetum*) are not located in the Rimavská basin area. The forest stands in some localities, e.g., Ťahan in the vicinity of the existing area of European importance (SKUEV0363), or Nad Opatovskou cestou, have a temporary character and tend to the habitat Ls3.2. Their classification into a specific habitat is not entirely clear and forest stands have the character of an interface between several habitats. However, vegetation variability is a natural property of nature and classification systems (vegetation units) often do not fully correspond to real vegetation. The final decision is made based on the prevailing features and must be substantiated by objective arguments. In these controversial cases, but also for all forest stands in the Rimavská basin, the key argument for non-inclusion in the Ls3.2 habitat is mainly the absence of xerothermic species of forest-steppe character, which require more basic soils with higher soil response values, and which are diagnostic for this habitat. e.g., *Lithospermum purpurocaeruleum* and *Dictamnus albus*.

Only the area Ťahan partially met these habitat conditions because the soil type is pararendzina. Nevertheless, the mentioned species did not occur on the site. In addition to these forest-steppe elements, some species were found abundantly in the model localities visited, which, on the contrary, were not recorded or rarely present in the investigated localities of the Rimavská basin, such as *Arum alpinum, Rhamnus cathartica, Melica uniflora, C. vitalba, P. mollis,* and *Convallaria majalis.* Although some species of the Ls3.2 habitat occur quite abundantly in the mapped localities, e.g., *C. michelii* or *F. rupicola,* the overall character of the vegetation is different.

The highest proportion of *Q. cerris* is most common in the forest stands of the Rimavská basin, which shifts the inclusion of vegetation in the Ls3.4 Pannonian-Balkanic turkey oak-seesile oak forests. In Italia the Turkey oak (*Q. cerris*) is widely distributed in localities, where it is the ecologically dominant oak on sandy and acidic soil [38]. Compared to Slovakia, it is identical on acidic soils (loess and sand).

In the territory is very common the undergrowth of the hornbeam (*C. betulus*), which is usually younger and lower than oak individuals. It probably spread to the stands due to the elimination of historical forms of management. In most of the examined localities it creates a continuous undergrowth, a shading level of the herbal layer, which as a result is very underdeveloped, very poor in species, often none. On the contrary, in model localities the hornbeam (*Carpinus* spp.) occurs only sporadically, the undergrowth is mainly formed by *A. campestre*, *Ulmus minor*, and the floor E_2_ is significant. While also experiencing range shifts, the minor European natives *Castanea sativa, S. torminalis*, and *Ulmus laevis* all considerably expand their range potential across climate change scenarios. Accompanied by *C. betulus,* with a stable range size, they hold the potential to substantially contribute to sustainably adapting European forests to climate change [39]. In this case, they play an important role in forest stands.

In most of the investigated localities was the geological subsoil of the river terrace, which was observable in the terrain mainly by the presence of rounds (stones formed into oval shapes during transport in the riverbed). The Teriakovce locality, the part near Maginhrad, was geologically significantly different. Andesite was the subsoil here. As a result, there were more skeletal soils at the site, and probably there was also a higher proportion of *Q. petraea*. Forests with a higher proportion of *Q. petraea* and mesophilic vegetation were mapped as the Ls2.1 habitat. A similarly high proportion of *Q. petraea* was in some parts of the Bretka locality. In most localities, the soil type was luvisol, developed on the loess cover of river terraces. In the case of flat relief, the soils were glazed, and the vegetation indicated seasonal wetting. In such habitats, *Q. robur* predominated, as well as on the bases of slopes in contact with the alluvium of current watercourses. At these slopes, the soil was more clayey, much darker, and with a polyhedral structure. These habitats could mostly be mapped as the Ls2.2 habitat. If more acidophilic taxa were present in these habitats, *F. alnus* was especially common, the classification tends to be Ls3.3. However, the character of the Ls3.3 habitat in the mapped localities is not clear and distinct, but transient and marginal. On slopes or convex relief shapes that did not show signs of gleystation, the luvisol was significantly lighter and clayey, probably because of more intense ilimerization (faster movement of soil water on convex relief and slopes than on plateaus where water stagnates). Such habitats are dominated by *Q. cerris,* and the stands have been mapped as Ls3.4 habitat. In this way, it is possible to define in a broader sense the basic principles of habitat and vegetation variability. However, the individual localities showed specific differences, such as the Ťahan locality, where they were a soil type of pararendzins.

As mentioned in the introduction, oak forests in Slovakia, but in temperate Europe in general, have changed significantly in recent decades due to changes in the elimination of historical forms of management and escalating climate change or eutrophication. The most significant element affecting the vegetation of the mapped localities is the mentioned extension of *Carpinus* spp. in the form of a continuous undergrowth, shading the level of the herbal layer. Because the plants typical of oak forests are light-loving, many are unable to survive this change. Vegetation in undergrowth *Carpinus* spp. is very poor in species, and there are mostly taxa tolerant of shading and indicating higher humidity. A good example is the location Mestský les, which is almost entirely overgrown with hornbeam. In places where there is undergrowth of *Carpinus* spp. the vegetation was immediately different, varied in species, and typical of oak forests. One of the historical forms of forest use is the excavation of litter, which also significantly affected the species composition. This was very probably the case in several places of the investigated localities, especially in the locality of Kružno, Mlynárka. The diversity and the ecological status of the oak forests in Bulgaria highlight the urgent need for their protection. They are in poor condition mostly due to the millennial tradition of fellings, coppicing, burning, and grazing. Further research is required to investigate their diversity, ecology, and dynamics—to gain the knowledge needed for their conservation, management, and restoration [40]. For example, the compositional diversity of Turkish oak woodland vegetation is extensive and reflects the biogeographical subdivision of the country. However, the ecology of several oak species and their ecosystems is still little known [41]. In Slovakia, the ecology of oak trees is very well developed. At present, little attention is paid to it, especially in the context of habitat assessment. An example is the absence of case studies—examples of the habitat we studied.

The localities were often dominated by a species of *L. luzuloides*, which is a clear indicator of litter excavation. Today, these species are not found in these places at all, and almost nothing grows in the herbaceous layer. In the Bretka locality, miliers were left in several places to produce charcoal. Near the village, right in the valley of the Muráň river, there was once a smelter for processing iron ore, where coal was used. According to the latest research [42,43], it can be stated that it was in the Slovak Karst that the massive production of charcoal for the metallurgical industry in the 18th century led to a change in the tree composition of forests. The use of forests related to coal production was mainly supported by *Quercus* spp. The subsequent decline until the cessation of production led to the spread of *Carpinus* spp. Similar findings are reported from other parts of Europe, where forests have served as the main source of energy for the charcoal industry.

For the detailed capture of landscape dynamics, manifestations of afforestation and deforestation of small areas, it is appropriate to use the methodology based on CORINE Land Cover data and methodology set up on satellite and aerial images interpretation, on detailed land cover interpretation (1:10,000) for the local case studies, as well as on the results of field research and forestry databases [44].

## 5. Conclusions

Based on our findings, we can confirm that the habitat of European importance 91I0* Euro-Siberian steppic woods with *Quercus* spp. is located in its subtypes Ls3.3 and Ls3.5.2—part B. There was an occurrence of subtype Ls3.2, to which the greatest attention was paid, but we did not manage to document in the area of the Lučenská and Rimavská basins, which does not mean that it is not located in this area of the south of Central Slovakia. In 2022, the SNC SR proposed other selected areas in this locality, where the monitoring and tracing of this habitat will continue. Identification of the Ls3.2 habitat is relatively difficult in the field. In some localities the stands tended to the given habitat, but their inclusion in it was not possible mainly due to the absence of xerothermic species of forest-steppe character and high representation of *Q. cerris*, due to which they were mapped as habitat 91M0 Pannonian-Balkan cereal forests. In the next part of the work, after comparing the typological map with the map of habitats created by us, we found out that forest types in the locality of the area of European importance Uderinky correspond to the present species of habitats. The converter from forest types to habitats based on the typological map used by SNC SR has proven itself in this case. Some boundaries do not match at all, which may be due to the subjectivity of the mapper. In research, it is important to build on the EU’s 2023 biodiversity strategy as it presents a comprehensive plan to protect nature and reverse the degradation of ecosystems as a priority for restoration, resilience to climate change, forest fires, including the protection of wildlife.

As part of the sequence of preventing habitat degradation, we consider it essential to implement in the near future (i) status and development surveys (favorable, unfavorable status, and habitat restoration for the benefit of nature and humans), (ii) monitoring and evaluation (regular monitoring of identified sites through changes in population ecology, which will provide detailed data on the species composition and quantitative proportions of species on the sites), and (iii) management planning (when restoring, use only gentle restoration methods that meet the criteria of close to nature forest management, ensure the best possible compliance of the woody composition with the natural woody composition on the site, and remove non-native species).

## Figures and Tables

**Figure 1 biology-12-00910-f001:**
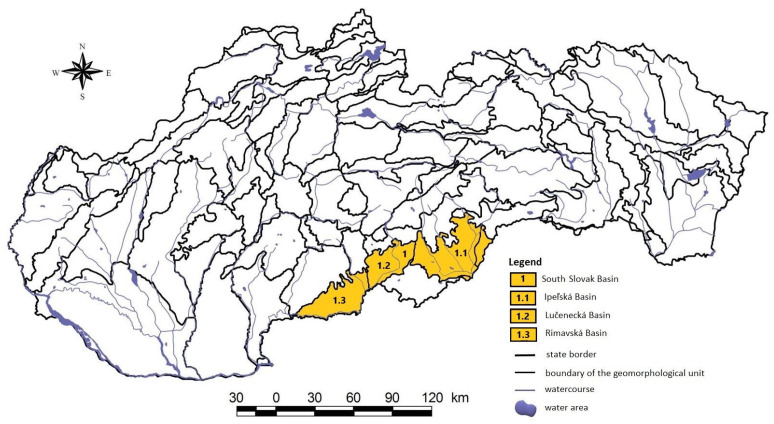
Location of study area on the South Slovak basin.

**Figure 2 biology-12-00910-f002:**
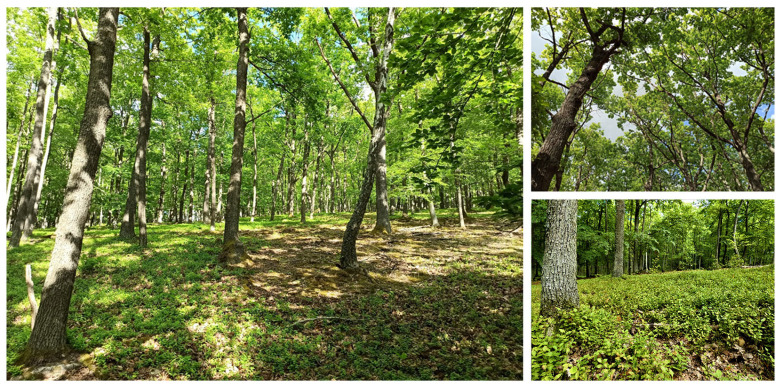
Area of European Importance (SKUEV0957) with an occurrence of *Vaccinium myrtillus*.

**Figure 3 biology-12-00910-f003:**
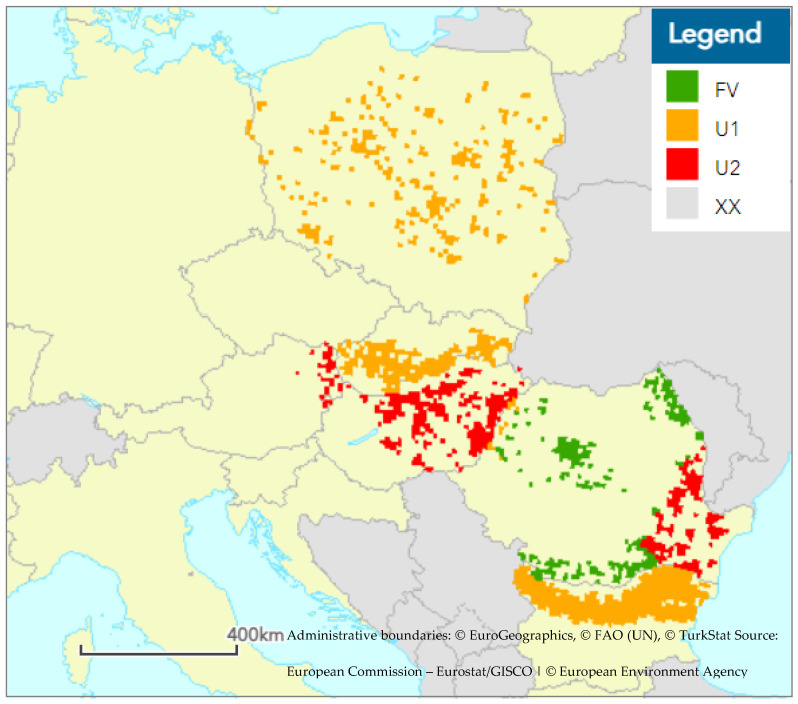
Habitat conservation status 91I0* for each biogeographical region (map shows assessments for 2013–2018 as reported by EU Member State). Legend: FV—Favourable, U1—Unfavourable–Inadequate, U2—Unfavourable–Bad, XX—Unknown [11].

**Figure 4 biology-12-00910-f004:**
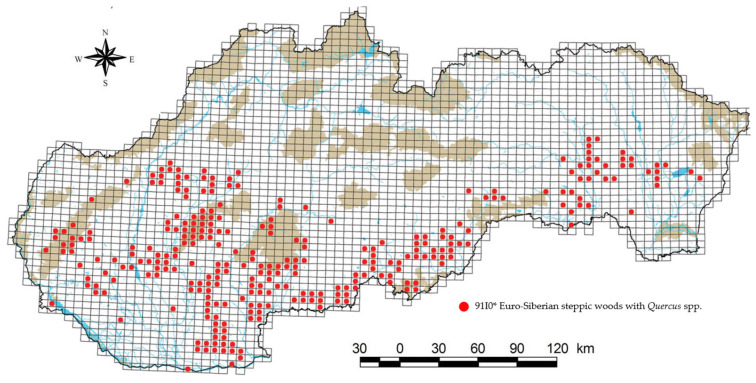
Map of habitat distribution within the Slovak Republic [30].

**Figure 5 biology-12-00910-f005:**
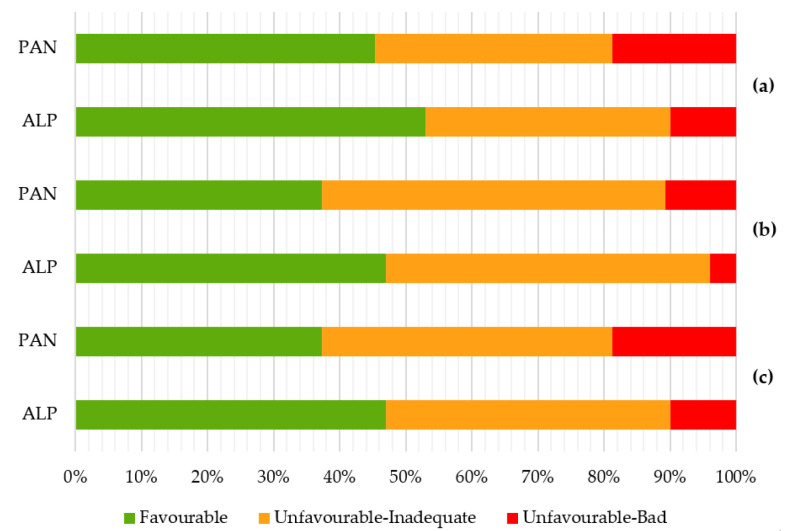
Habitat quality 91I0* in Slovak Republic: (**a**) status of habitat protection in the bioregion, (**b**) prospects of the habitat in the bioregion, and (**c**) quality of the habitat in the bioregion [27].

**Figure 6 biology-12-00910-f006:**
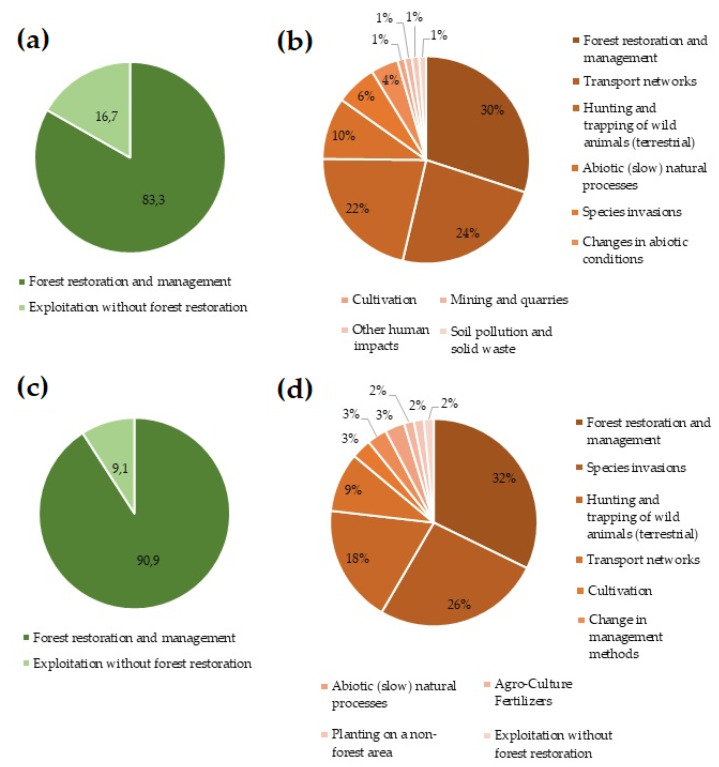
Frequency of positive and negative impacts: (**a**) frequency of positive impacts on the Alpine bioregion, (**b**) frequency of negative impacts on the Alpine bioregion, (**c**) frequency of positive impacts on the Pannonian bioregion, (**d**) frequency of negative impacts on the Pannonian bioregion.

**Figure 7 biology-12-00910-f007:**
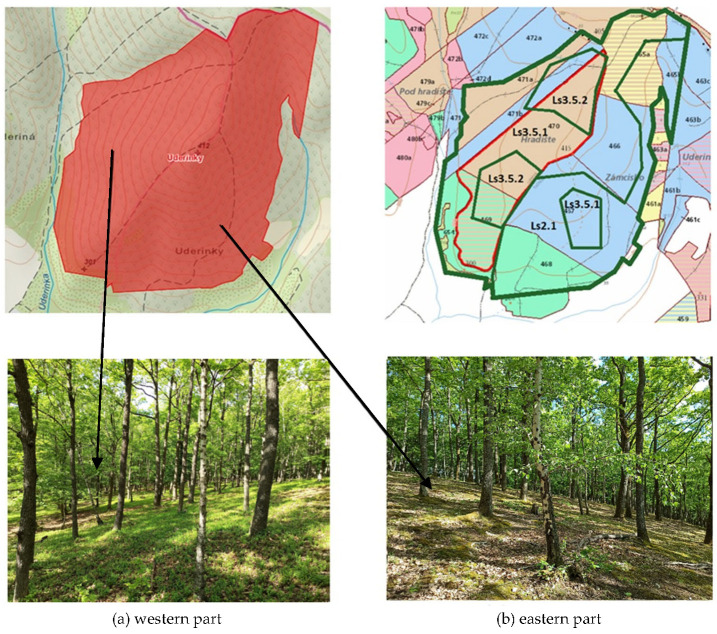
Habitat Ls3.5.2 Thermophilous and supra-Mediterranean oak woods for dust and sand [32,33].

**Table 1 biology-12-00910-t001:** Current selection: 2013–2018, Forests, 91I0* Euro-Siberian steppic woods with *Quercus* spp. [11].

Member States Reports	Bioregion	Range (km^2^): Status (% MS)	Area (km^2^): Status (% MS)	Status of Structure and Functions Including Typical Species	Status of Future Prospects	Conservation Status—Current Period	Distribution Area (km^2^): MS Region (%)
Bulgaria	Alpine	29.18	16.04	FV	U1	U1	12.61
	Black Sea	100	100	FV	U1	U1	100
	Continental	22.18	60.44	FV	U1	U1	34.34
Slovakia	Alpine	70.82	83.96	U1	U1	U1	87.39
	Pannonian	18.12	7.19	U1	U1	U1	25.74
Austria	Continental	1.95	0.47	FV	U2	U2	2.72
Czechia	Continental	14.91	1.93	U1	U1	U2	15.91
	Pannonian	13.52	17.30	U1	U1	U2	6.21
Poland	Continental	32.48	4.90	U1	U1	U1	25.70
Romania	Continental	28.48	32.26	FV	FV	FV	21.34
	Pannonian	15.77	37.40	FV	FV	U1	4.44
	Steppic	100	100	U1	FV	U1	100
Hungaria	Pannonian	52.59	38.12	U1	U1	U2	63.61

## Data Availability

Not applicable.

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
