# Peer review of "Endangered Forest Communities in Central Europe: Mapping Current and Potential Distributions of Euro-Siberian Steppic Woods with Quercus spp. in South Slovak Basin"

_biology, 2023, doi:10.3390/biology12070910_

Round 1

Reviewer 1 Report

Congratulations on the work done.

Undoubtedly, it is an applied botanical work of great interest for the proper conservation of this priority habitat.

I only missed vegetation maps of the 8 environments described in the results section (maybe you could add them as supplementary material?) and the phytosociological tables (maybe it would be interesting to show them in supplementary material?).

Finally, I have only detected some issues of format or content which I have highlighted in the attached document.

Author Response

Thank you for your valuable advice and comments. All comments of the review have been supplemented in the text. They are highlighted in green.

We added vegetation maps and phytosociological records to the appendix due to their large scale and detailed processing. The maps would not be clear and readable because of the scale. 

Reviewer 2 Report

Of course, biodiversity conservation is a critical task. Forest communities, particularly oak forests, are complex ecosystems with high biodiversity that must be protected. This article includes information about Slovakia's southern oak forests that require preservation. However, there are numerous critical remarks and comments on the article.

After carefully reading the manuscript, I never understood the novelty and scientific significance of this study. The manuscript resembles a technical report on the work completed for environmental organisations more than it does a scientific article. The article is mostly descriptive. This paper contains no results that might be relevant for advancing biological or forest science. The research approach is rather cursory, and the authors frequently make a reference to the reader becoming familiar with research methodologies independently by reading literary sources. However, the majority of these methodological publications are written in Slovak, and I doubt that most scientists (who will read this article) are native Slovak speakers. There may be some interest in Subsections 3.1 and 3.2. However, once again, because of the inadequate methodological section, it is unclear what the methodological foundation for determining the state of oak forests and the principles of combining various by type and degree of factor effects (anthropogenic and biotic) are.

Only environmental organisations may find the site descriptions (subsection 3.3) useful for organising conservation efforts. These site descriptions are not particularly interesting from a scientific standpoint. Additionally, it strikes me as odd that when characterising forest ecosystems, no features (!?) of the forests themselves are presented. At least the following characteristics. What is the average age of the stands? Average height of the trees? Mean DBH? The renewable potential of these forests? The amount of undergrowth under the canopy? What is the ratio between small and large undergrowth? What is the state of the undergrowth? Not to mention the absence of data on genetic diversity and genetic distance between these populations.

In my opinion, the manuscript does not add anything new to knowledge on this subject.

Author Response

We value the opinions and comments of the reviewer. Unfortunately, we have to disagree. We think that our local study is highly relevant and beneficial not only for the scientific society, but also for the general lay public, who will understand the clear goal and significance of the authors' work.

The article clearly defines the goals and methods that were implemented through field research. The benefit of the study is the identification of a priority habitat that has not yet been mapped in the selected area.

Almost no constructive or feasible proposal was made.

The reviews of other reviewers agree with our objection.

Reviewer 3 Report

1. The words of "Habitat 01I0","SKUEV0957", "steppic " should be explain what is the meaning in the abstract part.

2. The research aim of the manuscript was not unclear.

3. Line 180-185: The research work results divide three subsections description is not proper in results.

4. What is the FV, U1 and U2 evaluation criterion?

5. Line 261-263: How to judge the proper management and improper management?

6. The discussion of the manuscript logical relationship should be adjust.

7. For ease of following, please explain the meaning of ALP, BLS, CON, etc. in Table   

  1?

8. How many years into the future is the situation expressed in 3.1? What are the criteria for measuring how many years?

9. How are the percentages of positive and negative impacts in Figure 6 calculated?

Author Response

I appreciate all the reviewer's comments. Thank you for the valuable comments that we have processed into our text. They are highlighted in green.

Answer to question number 8 and 9: Assessment of habitat status at this level is based on the evaluation of sub-parameters: a) The quality of the biotope on the site and b) Biotope prospects on the site. For each parameter and status category, a percentage value is determined, while the sum of the different values states for each parameter must be 100% (e.g. population quality at a permanent monitoring site is good 30%, unsatisfactory 40% and bad 30%). The evaluation of these parameters will be carried out by the mapper during the field visit according to the established methodology for each habitat separately or by an expert estimate of the mapper who determines the status categories.

Round 2

Reviewer 2 Report

Because the paper was submitted to a high-ranking scientific (!) journal, it should, first and foremost, include useful scientific content and new scientific findings. This paper contains no results that might be relevant for advancing biological or forest science. The manuscript resembles a technical report on the work completed for environmental organisations more than a scientific article. The article is mostly descriptive.

The first section of the article is a review (i.e., a restatement of previously reported facts), while the second section is made up of descriptions of various forest habitats that are neither of particular scientific interest nor provide anything new to the field of forest or biological research.

Almost no constructive or feasible proposal was made….

Perhaps there were technical issues with the review submission system (not all the review content was forwarded to the authors). So let me repeat. Because your own results are only offered as descriptions of forest habitats, it is odd that descriptions of forest communities (!) lack features of the forests themselves. At least the following characteristics. What is the average age of the stands? Average height of the trees? Mean DBH? The renewable potential of these forests? The amount of undergrowth under the canopy? What is the ratio between small and large undergrowth? What is the state of the undergrowth? Not to mention the absence of data on genetic diversity and genetic distance between these populations. However, even if these data are included in the paper, they will not raise it to the level of a research article. Because, and I'll say it again, descriptions of forest habitats do not constitute a significant scientific finding.

In my opinion, the manuscript does not add anything new to knowledge on this subject.

Author Response

Thank you for your comment. Unfortunately, I do not identify with him. Two reviewers stated that undoubtedly, it is an applied botanical work of great interest for the proper conservation of this priority habitat.

We think our study meets the requirements of special issue. Respectively, it takes into account preliminary but significant results from the area ecological restoration of plant community.

Reviewer 3 Report

The manuscript is recommended for publication.

Author Response

Thank you for your comment and recommendation for publication.